# Associations of Serum Retinol and α-Tocopherol Levels with Uric Acid Concentrations: Analysis of a Population-Based, Nationally Representative Sample

**DOI:** 10.3390/nu12061797

**Published:** 2020-06-17

**Authors:** Yunkyung Kim, Jung Hee Choi, Jihun Kang, Geun-Tae Kim, Seung-Geun Lee

**Affiliations:** 1Division of Rheumatology, Department of Internal Medicine, Kosin University Gospel Hospital, Kosin University College of Medicine, Busan 49104, Korea; efmsungmo@hanmail.net (Y.K.); gtah@hanmail.net (G.-T.K.); 2Department of Anesthesiology and Pain Medicine, Chungbuk National University Hospital, Chungbuk National University College of Medicine, Cheongju 28644, Korea; choijh0612@gmail.com; 3Department of Family Medicine, Kosin University Gospel Hospital, Kosin University College of Medicine, Busan 49104, Korea; 4Central Institute for Medical Research, Kosin University Gospel Hospital, Busan 49104, Korea; 5Division of Rheumatology, Department of Internal Medicine, Pusan National University School of Medicine & Pusan National University Hospital, Busan 49241, Korea; sglee@pnuh.co.kr

**Keywords:** retinol, α-tocopherol, uric acid, hyperuricemia, gout, KNHANES

## Abstract

The effects of serum retinol and α-tocopherol on serum uric acid levels have not been established, especially in Asian people. This study evaluated the independent associations of retinol and α-tocopherol with serum uric acid levels in the Korean population. We included 6023 participants aged ≥ 19 years from the Korean National Health and Nutrition Examination Survey (KNHANES). Serum retinol and α-tocopherol levels were divided into quintiles, and a multivariate linear regression model was used to evaluate the association of serum retinol and α-tocopherol levels with uric acid concentration. Additionally, we used multivariate logistic regression to examine the relationships between the levels of these micronutrients and hyperuricemia. Serum retinol levels were positively associated with uric acid concentrations in a dose-dependent fashion in both sexes (*p*_trend_ < 0.001); the difference in serum uric acid levels between the highest and lowest quintiles of retinol levels was 0.57 mg/dL in men and 0.54 mg/dL in women. In the multivariable logistic model, the hyperuricemia risk increased linearly with the increase in serum retinol level, regardless of sex (*p*_trend_ < 0.001). Although the serum α-tocopherol level appeared to be significantly associated with increased uric acid levels, this association was nullified after adjusting for serum retinol levels. Serum retinol levels were positively associated with serum uric acid levels and hyperuricemia in a dose-response fashion. Maintaining serum retinol concentrations under sub-toxic levels might be necessary to prevent hyperuricemia-related adverse health outcomes.

## 1. Introduction

Hyperuricemia is not only a main risk factor for gout [1] but is also an independent determinant of type 2 diabetes [2], hypertension [3], metabolic syndrome [4], and chronic kidney disease [5]. Elevated serum uric acid is reported to be associated with an increased risk of non-dipping blood pressure [6], myocardial infarction [7], and cardiovascular mortality [7]. Serum uric acid levels may increase due to heavy alcohol consumption [8], high purine diet [9], impaired renal or extra-renal excretion [10], and increased cell turnover [11]. The prevalence of hyperuricemia is increasing worldwide. Approximately 21% of the US population had hyperuricemia in the 2007–2008 National Health and Nutrition Examination Survey (NHANES), a rate 3.2% higher than that in the 1988–1994 NHANES [12]; meanwhile, a 4.3% increase in prevalence was observed in Italy within 4 years (2005–2009) [13]. In Korea, although the prevalence of hyperuricemia was 11.4%, hyperuricemia was especially prevalent in young adults [14].

Vitamin A is an essential component for fetal growth, vision, and immune response [15], and deficiency of this micronutrient is associated with vision problems [16] and an increased risk of infection [17]. However, in developed countries where the consumption of vitamin A exceeds the Recommended Dietary Allowance (RDA) in a significant number of individuals [18], the toxic or sub-toxic effects of excessive vitamin A are perceived as a public health issue [19]. Regarding serum uric acid levels, synthetic vitamin A used for the treatment of acne was reported to have negative impacts on the development of hyperuricemia [20] and gout [21]. In addition, two population-based studies in the US reported that serum retinol concentration was independently associated with hyperuricemia [22,23], and a study on adolescents in the US also showed that uric acid levels were elevated in association with serum retinol and retinyl ester concentrations [24].

As an antioxidant and anti-inflammatory nutrient, α-tocopherol protects cells from free-radical damage [15] and is associated with a decreased risk of prostate cancer [25]. However, vitamin E supplementation can increase the risk of brain hemorrhage as well as overall mortality [26,27]. Limited studies have been conducted on the effects of α-tocopherol on serum uric acid levels, and the results were inconsistent. An experimental study on rats showed that vitamin E was associated with decreased serum uric acid levels [28]. In contrast, a study on chronic kidney disease patients observed no significant association between serum vitamin E and uric acid levels [29].

Despite previous research, the effects of serum retinol and α-tocopherol on uric acid concentrations have not been established, especially in Asian populations. Furthermore, compared to studies evaluating these micronutrients using food frequency questionnaires, relatively few studies have assessed serum retinol and α-tocopherol levels. Therefore, this study aimed to evaluate the association of serum retinol and α-tocopherol levels with hyperuricemia in the general Korean population.

## 2. Materials and Methods

### 2.1. Study Population

This study used data from the Korea National Health and Nutritional Examination Survey (KNHANES). The survey is described in detail elsewhere [30]. Briefly, the KNHANES is a nationally representative study that collects data on non-institutionalized South Korean citizens that is conducted annually by the Korean Center for Disease Control and Prevention (KCDC, Chongzhou, Korea). This survey comprises a health interview, health examination, and nutritional survey and uses stratified, multi-stage, clustered probability sampling to obtain a nationally representative sample.

We used data from KNHANES 2016–2018, which included measurements of serum retinol, α-tocopherol, and uric acid levels. Among 31,689 eligible individuals, 24,269 participated in the survey, corresponding to a response rate of 76.6%; of these respondents, serum retinol and α-tocopherol levels were measured in 7189 randomly selected participants. Among them, 6424 participants aged ≥ 19 years with available information on retinol and α-tocopherol concentrations and serum uric acid levels were identified. Additionally, 401 participants were excluded for the following reasons: previous diagnosis of renal cell carcinoma (*n* = 10); pregnancy (*n* = 20); glomerular filtration rate (GFR) < 10 mL/min/1.73 m^2^ (*n* = 5); outliers of serum retinol (*n* = 1 [2.27 mg/L]) or α-tocopherol (*n* = 2 [95 and 140 mg/L, respectively]) values; or missing data on anthropometric variables (body mass index [BMI, *n* = 12], blood pressure [*n* = 13]), education level (*n* = 25), alcohol consumption (*n* = 44), smoking status (*n* = 63), physical activity (*n* = 202), and high sensitivity C-reactive protein levels (hs-CRP, *n* = 4). Finally, the analysis included 6023 participants.

The study protocol was approved by the Kosin University Gospel Hospital Institutional Review Board (IRB, No. KUGH 2020-IRB 2020-03-053) and complied with the tenets of the Declaration of Helsinki. All study procedures also complied with Strengthening the Reporting of Observational studies in Epidemiology (STROBE) guidelines, and written informed consent was obtained from all participants.

### 2.2. Data Collection and Measurement

Trained research assistants collected information on sociodemographic variables (age, sex, area of residence, and education) during face-to-face interviews. Data regarding health behaviors (alcohol consumption, smoking status, and physical activity) were obtained from self-administered questionnaires. Residential areas were categorized as rural or urban, and education level was categorized as elementary school, middle school, high school, or university or above. Alcohol consumption was defined as ≥7 and ≥5 drinks on an occasion in men and women, respectively, and further categorized into no drinks, less than once per week, and more than or equal to once per week. Based on the World Health Organization (WHO) criteria, smoking status was categorized into non, past, and current smokers. Participants who engaged in moderate physical activity for ≥150 min per week or in vigorous activity ≥75 min per week were defined as physically active according to WHO global recommendations [31] and were dichotomously categorized.

Trained medical assistants assessed weight, height, and blood pressure in the specially designed mobile center for physical examinations. Height (kg, SECA, Hamburg, Germany) and weight (m, GL-6000-20, Daegu, Korea) were measured while participants wore light clothing without shoes; BMI was calculated as the weight divided by the square of the height. Blood pressure was measured thrice using a mercury sphygmomanometer (Baum, Sidney, OH, USA) and stethoscope (3M, St. Paul, MN, USA) after participants had rested for 5 min. The mean of the latter two blood pressure values was used, regardless of the first measured value.

Venous blood samples were drawn through venipuncture using a vacutainer needle and tubes. Once serum was separated, laboratory tests were performed to measure serum retinol, α-tocopherol, uric acid creatinine, and CRP levels. Serum retinol and α-tocopherol levels were measured by liquid chromatography–flame ionization (Agilent1200, Agilent, Santa Clara, CA, USA) with reference ranges of 0.30–0.70 mg/L and 5.00–20.00 mg/L, respectively. Kinetic Jaffe assays (Hitachi automatic analyzer 7600–210, Tokyo, Japan) were used to measure serum creatinine levels, and the GFR was calculated using the Chronic Kidney Disease Epidemiology Collaboration equation. Immunoturbidimetry was used to measure hs-CRP levels (Cobas, Roche, Germany). Serum uric acid levels were assessed by colorimetry with a uricase–catalase system (Hitachi automatic analyzer 7600–210, Tokyo, Japan). Hyperuricemia was defined as a serum urate level >7.0 mg/dL in men and >6.0 mg/dL in women.

### 2.3. Statistical Analysis

The KNHANES constructed a sampling plan using a multistage clustered probability design to obtain nationally representative data. Based on census data, sample weights were generated to address the complex survey design, post-stratification data, and non-responders. Therefore, all analyses used the complex survey design and sample weights.

We compared the general characteristics of the study participants according to hyperuricemia status. For normally distributed continuous variables, chi-squared tests were performed to assess normally distributed continuous variables and categorical variables. As the hs-CRP values were not normally distributed, we performed comparisons using Mann–Whitney U-tests.

Because fortified foods and dietary supplements commonly contain both vitamins A and E, we used Pearson correlation coefficients to explore the correlations between serum retinol and α-tocopherol levels. In addition, we used a general linear regression model to evaluate the association of retinol and α-tocopherol concentrations with serum uric acid levels. We included variables with *p* < 0.1 in the univariate analysis or with significant associations with uric acid (as reported in previous studies) as covariates. Model 1 was adjusted for age; Model 2 was additionally adjusted for BMI and GFR; Model 3 was additionally adjusted for area of residence, education level, systolic and diastolic blood pressure, alcohol consumption, smoking status, and physical activity; Model 4 was additionally and mutually adjusted for serum retinol and α-tocopherol levels. Quintile distributions of serum retinol and α-tocopherol levels were inserted in the analysis models as continuous variables to estimate the dose-dependent relationship between serum micronutrient and uric acid levels, with the lowest quintile set as the reference value for comparisons. In addition, we conducted a subgroup analysis of the association between dietary retinol intake and serum uric acid levels among participants with dietary information obtained using a 24-h dietary recall method.

We conducted multivariate logistic analysis to estimate the risk of hyperuricemia in relation to retinol and α-tocopherol levels after adjusting for the mentioned covariates. The *p*-values for trends were calculated to examine whether the risk of hyperuricemia increased with higher serum micronutrient levels.

In addition, a sensitivity analysis was conducted to assess whether the use of dietary supplements modified the association between micronutrients and hyperuricemia. The effects of stratification were determined by inserting an interaction term of dietary supplement use in the final multivariate logistic analysis.

All tests were two-tailed, and *p*-values < 0.05 were considered statistically significant. All analyses were performed using IBM SPSS Statistics for Windows, version 24.0 (IBM Corp., Armonk, NY, USA).

## 3. Results

### 3.1. General Characteristics of Study Participants

The prevalence of hyperuricemia was 18.5% in men and 5.7% in women. The mean serum retinol and α-tocopherol levels were higher in hyperuricemia participants than in non-hyperuricemia participants. While men with hyperuricemia were younger, the mean age of women with hyperuricemia was higher than that of women without hyperuricemia. Education level, BMI, systolic and diastolic blood pressure, and hs-CRP levels were higher in the hyperuricemia group than in the non-hyperuricemia group for both sexes. Female participants with hyperuricemia were more likely to be current smokers and to have lower GFR than women without hyperuricemia (Table 1).

### 3.2. Correlation between Serum Retinol and α-Tocopherol

The inter-relationships between serum retinol and α-tocopherol levels are presented in Figure 1. Serum retinol was positively correlated with α-tocopherol (*r*^2^ = 0.123, *p* < 0.001 in men and *r*^2^ = 0.136, *p* < 0.001 in women).

### 3.3. Association of Serum Retinol and α-Tocopherol Concentration with Uric Acid Levels

Uric acid levels steadily increased with higher serum retinol levels (from 5.61 mg/dL in the lowest retinol quintile [Q1] to 6.27 mg/dL in the highest quintile [Q5] in men and from 4.25 mg/dL in Q1 to 4.96 mg/dL in Q5 in women) after adjusting for age (*p*_trend_ < 0.001 in both sexes). In fully adjusted analyses, the difference in serum uric acid levels across quintiles of retinol was slightly reduced, from 0.66 mg/dL to 0.57 mg/dL in men and from 0.71 mg/dL to 0.54 mg/dL in women, but remained significant. Although serum α-tocopherol levels were positively associated with serum uric acid levels from Models 1 to 3, the association was not significant after adjusting for serum retinol levels (Table 2). However, there was no association between the dietary intake of retinol and serum uric acid level in both sexes (Appendix A).

### 3.4. Association between Serum Retinol and α-Tocopherol Levels and Hyperuricemia

In the multivariable logistic regression model, the risk of hyperuricemia increased with the increase in serum retinol and α-tocopherol levels (*p*_trend_ < 0.001 in both sexes). The odds ratio (OR) for hyperuricemia in Q5, compared to Q1 of retinol levels, was 3.37 in men and 3.95 in women. The concentration of α-tocopherol showed a tendency to increase the risk of hyperuricemia in the age-adjusted analysis; however, the association disappeared after adjusting for serum retinol levels (Figure 2).

### 3.5. Interactive Effects of Dietary Supplements and Serum Retinol and α-Tocopherol on Hyperuricemia

Our analysis exploring the potential interactive effect of dietary supplementation on the association between micronutrients and hyperuricemia showed no significant interaction, except for serum retinol levels in women (*p*
_interaction_ = 0.012). While the magnitude of the association between serum retinol levels and the risk of hyperuricemia increased in women taking dietary supplements, the strength of the association decreased in men taking dietary supplements (Table 3).

## 4. Discussion

This nationally representative study showed that serum retinol levels were positively associated with elevated serum uric acid concentrations in a dose-responsive fashion. In addition, the risk of hyperuricemia increased linearly with increasing serum retinol levels. Serum retinol and α-tocopherol levels showed significant correlations with each other, and the serum retinol level was a confounding factor affecting the relationship between α-tocopherol and serum uric acid levels. The major strengths of the present study are (1) its nationally representative sample that provides sufficient statistical power and representativeness; (2) robust adjustment for potential compounders related to serum uric acid levels; and (3) reliable measurements of micronutrients using participants’ serum samples.

The results of the current study suggested that serum retinol levels had a dose-dependent association with serum uric acid levels and the risk of hyperuricemia, consistent with the findings of two studies conducted in the US [22,23] which reported that serum uric acid levels increased with increasing retinol concentrations. The differences in uric acid levels between the highest and lowest quintiles of retinol levels were 0.70 and 0.48 mg/dL in the 1998–1994 [22] and 2001–2006 NHANESs, respectively [23]. The magnitude of difference in uric acid levels in our study was 0.57 mg/dL in men and 0.54 mg/dL in women; these values are between those reported by the two previous studies in the US. When we translated our observed findings to clinical practice, the difference in uric acid levels between the top and bottom quintiles of retinol concentrations was higher than the increase in uric acid concentrations among individuals who consumed ≥1 shot of liquor/day (0.26 mg/dL) and comparable to the increase in uric acid levels among individuals drinking ≥1 beer/day (0.42 mg/dL) [8]; however, direct comparisons are limited due to heterogeneities in the study design and measurement methods. Furthermore, a study on adolescents in the US reported positive associations of serum retinol and retinyl ester levels with uric acid concentration, which partially support our findings [24].

Hyperuricemia increased with increasing serum retinol levels, indicating that the consumption of vitamin A exceeding the recommended values from fortified foods and supplements could be responsible for adverse health outcomes related to hyperuricemia, including not only gout [1] but also hypertension [3], metabolic syndrome [4], chronic kidney disease [5], and myocardial infarction [7]. Previous studies have shown that high serum retinol levels had a negative effect on the development of osteoporosis and hip fractures [32,33]. In addition, the dietary supplementation of vitamin A significantly increased mortality [26]. Thus, dietary modification to maintain serum vitamin A at adequate but not excessive levels and the individualized use of dietary supplements might be necessary to prevent vitamin A overconsumption, not only in individuals with these health conditions, but also in the general population.

Although the use of dietary supplements strengthened the association between serum retinol levels and the risk of hyperuricemia in women, this effect was not observed in men. It is difficult to explain the effect modification of dietary supplements on hyperuricemia observed only in women; however, the higher use of the supplements in women (49.1%) than in men (37.5%) might be related to this finding. In addition, varying compositions and types of supplements according to sex could play a role in the intensified association between retinol and uric acid levels only in women [34].

Regarding the effect of α-tocopherol levels on serum uric acid levels, this study did not show any beneficial effect of α-tocopherol on the risk of hyperuricemia. Although vitamin E supplementation was shown to exhibit significant uricosuric effects and to decrease serum uric acid levels in an experimental study on rats [28], another clinical study on chronic kidney patients did not report a significant association between vitamin E and serum uric acid levels [29]. Notably, before adjusting for serum retinol levels, vitamin E levels appeared to be significantly associated with increased serum uric acid levels. However, this significant association was nullified after adjusting for serum retinol levels. This finding suggested that retinol levels could be a significant compounding factor in the analysis of α-tocopherol levels and hyperuricemia. The concomitant fortification of vitamin A and vitamin E in foods and supplements might be a plausible mechanism underlying this compounding effect [35]. Furthermore, a sex-based difference was observed in this association. Only in males did the strength of the association between serum α-tocopherol and uric acid level disappear before adjusting for serum retinol levels. This could be hypothesized as follows: One possible explanation is that the presence of metabolic impairments in males due to smoking, increased alcohol consumption, and high blood pressure could have influenced serum uric acid levels. Another possible explanation is that estrogen’s protective role against hyperuricemia might be attributable to this mitigated degree of association observed only in males [36]. Nonetheless, further interventional studies are needed to elucidate the mechanisms underlying the observed findings.

The mechanism underlying the relationship between serum retinol levels and hyperuricemia is not yet fully understood. One possible explanation is that increased xanthine oxidase activity resulting from high serum retinol levels plays an important role in elevating serum uric acid levels. Xanthine oxidase converts retinol to retinoic acid [37] and catalyzes xanthine oxidation to produce uric acid [38]. Increased xanthine oxidase activity induced by high serum retinol levels could cause hyperuricemia by facilitating uric acid synthesis. In addition, other factors concurrently linked to high serum retinol levels and hyperuricemia, such as obesity and the intake of high-fat foods, might contribute to these two conditions [39]. However, the independent association between retinol levels and hyperuricemia remained after adjusting for these potential compounding factors in the analysis.

This study has several limitations. First, as the cross-sectional design could not address temporality issues, caution is needed when interpreting the results. In addition, although it is more likely that serum retinol levels affected serum uric acid levels, we could not exclude the possibility of reverse causality. Second, the use of uric acid-lowering medications, such as febuxostat, allopurinol, or rasburicase, was not evaluated due to data unavailability; hence, the rate of hyperuricemia might have been underestimated. However, given the prevalence of gout in South Korea (0.76%) [40], it seems unlikely that the use of these medications significantly affected our study results. Third, we could not include dietary factors other than alcohol consumption in the analysis due to the low response rate to the food intake frequency questionnaire. Fourth, other potential factors, such as serum vitamin D and parathyroid hormone (PTH) levels, were not included in the analysis because the KNHANES 2016–2018 did not measure these variables. Fifth, when compared with serum uric acid levels, serum retinol would not be a better indicator to monitor adverse health conditions, such as hypertension, chronic kidney disease, and metabolic syndrome, owing to cost effectiveness. Finally, because the liver is the main reservoir of vitamin A, serum retinol levels might not reflect excessive vitamin A states in the human body [41].

## 5. Conclusions

Serum retinol levels were positively associated with serum uric acid levels and hyperuricemia in a dose-response fashion. Maintaining serum retinol concentrations under sub-toxic levels might be an effective strategy to prevent adverse health outcomes related to hyperuricemia. Further studies are warranted to confirm and elucidate the observed associations.

## Figures and Tables

**Figure 1 nutrients-12-01797-f001:**
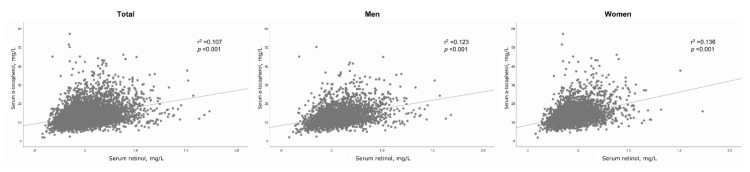
Correlation between serum retinol and α-tocopherol levels. Data are presented as scatter plots with lines of best fit.

**Figure 2 nutrients-12-01797-f002:**
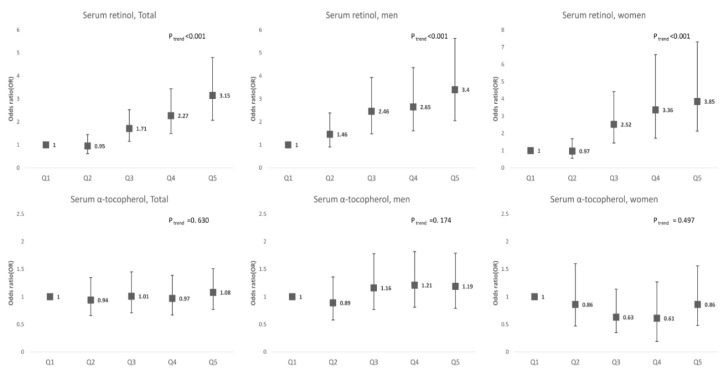
Association of serum retinol and α-tocopherol levels with hyperuricemia. Data are presented as odds ratio with 95% confidence intervals (error bars).

**Table 1 nutrients-12-01797-t001:** General characteristics of study population.

	Total			Men			Women		
	Hyperuricemia (*N* = 665)	No Hyperuricemia (*N* = 5358)	*p*-Value	Hyperuricemia (*N* = 472)	No Hyperuricemia (*N* = 2250)	*p*-Value	Hyperuricemia (*N* = 193)	No Hyperuricemia (*N* = 3108)	*p*-Value
Age, mean (SE)	42.4(0.7)	45.9(0.3)	<0.001	40.2 (0.7)	45.8 (0.4)	<0.001	49.7 (1.6)	46.0 (0.3)	0.022
Area of residence, % (SE)			0.619			0.838			0.318
Urban	85.6 (2.0)	86.3 (1.4)		86.3 (2.2)	86.6 (1.5)		83.2 (3.4)	86.1 (1.5)	
Rural	14.4 (2.0)	13.7 (1.4)		13.7 (2.2)	13.4 (1.5)		16.8 (3.4)	13.9 (1.5)	
Education, % (SE)			0.061			0.028			0.040
Elementary school	8.5 (1.1)	12.4 (0.5)		4.4 (0.8)	9.1 (0.6)		22.7 (3.3)	15.5 (0.7)	
Middle school	8.3 (1.1)	8.3 (0.4)		7.5 (1.3)	7.5 (0.6)		11.0 (2.2)	9.0 (0.5)	
High school	38.7 (2.4)	35.5 (0.8)		39.8 (2.8)	36.9 (1.2)		34.9 (4.0)	34.3 (1.0)	
University or above	44.5 (2.5)	43.7 (1.0)		48.2. (2.9)	46.6 (1.3)		31.5 (4.4)	41.2 (1.1)	
Smoking status *			0.001			0.413			0.016
Non-smokers	44.4 (2.3)	60.5 (0.7)		32.3 (2.6)	29.4 (1.1)		86.4 (2.7)	89.0 (0.7)	
Past smokers	23.8 (1.9)	18.5 (0.6)		29.8 (2.3)	33.1 (1.1)		2.9 (1.1)	5.1 (0.5)	
Current smokers	31.8 (2.1)	21.0 (0.7)		37.9 (2.6)	37.5 (1.3)		10.7 (2.6)	5.8 (0.5)	
Alcohol consumption, % (SE) ^†^			0.001			0.090			0.171
None	34.1 (2.1)	45.7 (0.8)		25.2 (2.3)	31.0 (1.1)		64.9 (3.9)	59.2 (1.1)	
<1/week	33.6 (2.3)	32.7 (0.8)		36.6 (2.8)	35.1 (1.2)		23.2 (3.8)	30.6 (1.0)	
≥1/week	32.3 (2.2)	21.6 (0.7)		38.2 (2.6)	34.0 (1.2)		11.9 (2.5)	10.2 (0.7)	
Physical activity, % (SE) ^‡^			0.368			0.897			0.657
No	49.4 (2.3)	51.5 (0.9)		48.3 (2.7)	47.9 (1.2)		53.0 (4.2)	54.8 (1.1)	
Yes	50.6 (2.3)	48.5 (0.9)		51.7 (2.7)	52.1 (1.2)		47.0 (4.2)	45.2 (1.1)	
Body mass index, % (SE)	26.2 (0.2)	23.6 (0.1)	<0.001	26.1 (0.2)	24.2 (0.1)	<0.001	26.3 (0.4)	23.1 (0.1)	<0.001
Systolic blood pressure, mmHg	121.6 (0.7)	116.4 (0.3)	<0.001	122.1 (0.7)	119.1 (0.4)	<0.001	120.1 (2.0)	114.0 (0.3)	0.002
Diastolic blood pressure, mmHg	79.9 (0.5)	75.6 (0.2)	<0.001	81.2 (0.5)	78.0 (0.3)	<0.001	75.3 (0.9)	73.4 (0.2)	0.037
GFR, mL/min/1.73m^2^	94.1 (0.9)	99.6 (0.3)	<0.001	95.4 (0.9)	96.8 (0.4)	0.161	89.7 (2.0)	102.1 (0.3)	<0.001
hs-CRP, mg/L									
Median (IQR)	0.8 (0.5,1.6)	0.5 (0.3,0.98)	<0.001	0.78 (0.48,1.53)	0.55 (0.35,1.04)	<0.001	0.93 (0.6,1.78)	0.46 (0.3,0.89)	<0.001
Uric acid, mg/dL	7.64 (0.04)	4.86 (0.02)	<0.001	7.89 (0.04)	5.50 (0.02)	<0.001	6.74 (0.05)	4.27 (0.02)	<0.001
Serum retinol, mg/L	0.61 (0.01)	0.51 (0.01)	<0.001	0.63 (0.01)	0.57 (0.01)	<0.001	0.54 (0.02)	0.46 (0.01)	<0.001
Serum α-tocopherol, mg/L	14.1 (0.3)	13.3 (0.1)	0.006	14.0 (0.3)	13.1 (0.1)	0.008	14.3 (0.4)	13.5 (0.1)	0.074

Data are presented as weighted percentages (standard error [SE]) or weighted means (SE) unless otherwise stated. *p*-values were calculated with the use of a chi-squared test for categorical variables, a Student’s *t*-test for continuous variables, and a Mann–Whitney U-test for non-normally distributed variables, respectively. Hyperuricemia was defined as serum urate level >7.0 mg/dL and >6.0 mg/dL in men and women, respectively. * Smoking status was categorized based on World Health Organization (WHO) criteria. ^†^ Alcohol consumption was defined as ≥7 drinks in men and ≥5 drinks in women on an occasion. ^‡^ Physical activity was defined as ≥150 min of moderate activity per day or ≥75 min of vigorous activity per day based on the WHO recommendation.

**Table 2 nutrients-12-01797-t002:** Association of serum retinol and α-tocopherol concentration with uric acid levels.

	Q1	Q2	Q3	Q4	Q5	*p*_trend_ *
Total						
Retinol						
Model 1 *	4.83 (4.76,4.91)	5.03 (4.95,5.11)	5.15 (5.07,5.22)	5.33 (5.25,5.41)	5.56 (5.47,5.65)	<0.001
Model 2 ^†^	4.93 (4.87,5.00)	5.06 (4.99,5.13)	5.16 (5.09,5.23)	5.30 (5.23,5.38)	5.47 (5.38,5.55)	<0.001
Model 3 ^‡^	4.90 (4.81,4.99)	5.04 (4.96,5.13)	5.17 (5.08,5.26)	5.31 (5.23,5.40)	5.49 (5.39,5.59)	<0.001
Model 4 ^§^	4.89 (4.80,4.99)	5.04 (4.95,5.13)	5.17 (5.08,5.26)	5.31 (5.23,5.40)	5.49 (5.39,5.59)	<0.001
α-tocopherol						
Model 1 *	5.04 (4.96,5.11)	5.1 (5.02,5.17)	5.17 (5.09.5.25)	5.30 (5.21,5.38)	5.31 (5.23,5.39)	<0.001
Model 2 ^†^	5.10 (5.02,5.17)	5.13 (5.07,5.20)	5.18 (5.11,5.25)	5.26 (5.18,5.34)	5.26 (5.18,5.33)	<0.001
Model 3 ^‡^	5.13 (5.04,5.22)	5.17 (5.08,5.25)	5.21 (5.12,5.30)	5.28 (5.19,5.38)	5.26 (5.17,5.36)	0.002
Model 4 ^§^	5.19 (5.10,5.28)	5.17 (5.09,5.25)	5.19 (5.10,5.28)	5.23 (5.13,5.32)	5.14 (5.05,5.24)	0.849
Men						
Retinol						
Model 1 *	5.61 (5.50,5.73)	5.79 (5.67,5.91)	6.01 (5.89,6.13)	6.09 (5.96,6.22)	6.27 (6.14,6.40)	<0.001
Model 2 ^†^	5.70 (5.59,5.80)	5.83 (5.71,.594)	6.02 (5.91,6.12)	6.03 (5.90,6.15)	6.19 (6.06,6.32)	<0.001
Model 3 ^‡^	5.67 (5.54,5.80)	5.85 (5.71,5.99)	6.05 (5.92,6.19)	6.08 (5.94,6.23)	6.24 (6.09,6.39)	<0.001
Model 4 ^§^	5.67 (5.53,5.80)	5.85 (5.71,5.99)	6.05 (5.92,6.19)	6.08 (5.94,6.23)	6.24 (6.09,6.40)	<0.001
α-tocopherol						
Model 1 *	5.78 (5.66,5.90)	5.84 (5.73,5.96)	5.98 (5.86,6.10)	6.09 (5.95,6.22)	6.05 (5.91,6.18)	<0.001
Model 2 ^†^	5.86 (5.75,5.98)	5.87 (5.77,5.98)	5.97 (5.86,6.08)	6.03 (5.90,6.17)	5.99 (5.86,6.11)	0.034
Model 3 ^‡^	5.90 (5.76,6.04)	5.91 (5.78,6.03)	5.99 (5.86,6.12)	6.06 (5.90,6.21)	6.00 (5.85,6.15)	0.097
Model 4 ^§^	6.00 (5.86,6.15)	5.93 (5.81,6.06)	6.00 (5.87,6.13)	6.02 (5.87,6.17)	5.90 (5.75,6.06)	0.549
Women						
Retinol						
Model 1 *	4.25 (4.19,4.30)	4.38 (4.30,4.45)	4.59 (4.49,4.70)	4.74 (4.60,4.87)	4.96 (4.77,5.14)	<0.001
Model 2 ^†^	4.28 (4.23,4.33)	4.37 (4.30,4.45)	4.56 (4.46,4.66)	4.68 (4.55,4.80)	4.83 (4.67,4.99)	<0.001
Model 3 ^‡^	4.35 (4.24,4.45)	4.44 (4.34,3.54)	4.63 (4.51,4.76)	4.75 (4.61,4.88)	4.89 (4.72,5.06)	<0.001
Model 4 ^§^	4.35 (4.25,4.45)	4.45 (4.34,4.55)	4.63 (4.51,4.76)	4.75 (4.61,4.88)	4.89 (4.72,5.07)	<0.001
α-tocopherol						
Model 1 *	4.30 (4.21,4.39)	4.36 (4.27,4.44)	4.38 (4.30,4.47)	4.48 (4.39,4.57)	4.56 (4.47,4.65)	<0.001
Model 2 ^†^	4.33 (4.25,4.41)	4.36 (4.28,4.45)	4.41 (4.33,4.49)	4.46 (4.37,4.54)	4.51 (4.43,4.60)	0.001
Model 3 ^‡^	4.45 (4.33,4.56)	4.50 (4.39,4.61)	4.53 (4.41,4.64)	4.57 (4.46,4.68)	4.61 (4.49,4.72)	0.006
Model 4 ^§^	4.46 (4.35,4.57)	4.48 (4.37,4.59)	4.48 (4.36,4.60)	4.49 (4.38,4.60)	4.47 (4.34,4.59)	0.996

Data are presented with mean (standard error). * *p*-values for trend were calculated using linear regression analysis by considering quintile distribution of serum retinol and α-tocopherol levels as continuous variables. ^†^ Model 1 adjusted for age; ^‡^ Model 2 additionally adjusted for body mass index (BMI) and glomerular filtration rate (GFR); ^§^ Model 3 additionally adjusted for residence, education, smoking status, alcohol consumption, physical activity, systolic and diastolic blood pressure, and log transformed high sensitivity C-reactive protein (hs-CRP); Model 4 additionally adjusted for serum retinol and α-tocopherol levels mutually.

**Table 3 nutrients-12-01797-t003:** Use of dietary supplements stratified association of serum retinol and α-tocopherol concentration with hyperuricemia ^‡^.

	Q1	Q2	Q3	Q4	Q5	*p*_trend_ *	*p* _interaction_ ^†^
Total							
Retinol							
Dietary supplements	Reference	0.72 (0.37,1.39)	1.17 (0.59,2.33)	1.91 (0.97,3.73)	2.24 (1.13,4.45)	<0.001	0.617
No dietary supplements	Reference	1.13 (0.66,1.96)	2.13 (1.26,3.61)	2.48 (1.42,4.31)	3.89 (2.25,6.74)	<0.001	
α-tocopherol							
Dietary supplements	Reference	1.02 (0.55,1.90)	1.01 (0.55,1.86)	0.71 (0.38,1.32)	1.29 (0.72,2.30)	0.622	0.295
No dietary supplements	Reference	0.90 (0.58,1.40)	1.04 (0.67,1.61)	1.18 (0.76,1.85)	1.00 (0.65,1.53)	0.565	
Men							
Retinol							
Dietary supplements	Reference	1.02 (0.38,2.70)	2.34 (0.91,6.00)	1.68 (0.65,4.43)	2.31 (0.90,5.95)	0.033	0.604
No dietary supplements	Reference	1.67 (0.96,2.92)	2.37 (1.37,4.10)	3.16 (1.78,5.62)	3.93 (2.16,7.14)	<0.001	
α-tocopherol							
Dietary supplements	Reference	0.67 (0.30,1.51)	0.98 (0.44,2.18)	0.98 (0.47,2.08)	1.31 (0.62,2.77)	0.247	0.764
No dietary supplements	Reference	0.96 (0.57,1.61)	1.24 (0.73,2.10)	1.33 (0.78,2.26)	1.09 (0.65,1.82)	0.373	
Women							
Retinol							
Dietary supplements	Reference	0.37 (0.15,0.96)	1.26 (0.53,3.00)	4.36 (1.87,10.15)	5.61 (2.06,15.32)	<0.001	0.012
No dietary supplements	Reference	1.46 (0.71,2.98)	3.77 (1.78,7.97)	2.19 (0.79,6.09)	3.45 (1.53,7.75)	0.001	
α-tocopherol							
Dietary supplements	Reference	0.85 (0.30,2.40)	0.61 (0.22,1.70)	0.28 (0.08,1.01)	0.66 (0.22,1.97)	0.255	0.618
No dietary supplements	Reference	0.79 (0.37,1.68)	0.64 (0.30,1.38)	0.82 (0.35,1.92)	0.90 (0.43,1.88)	0.837	

Data are presented with means (standard errors). * *p*-values for trend were calculated using logistic regression analysis by considering quintile distribution of serum retinol and α-tocopherol levels as continuous variables. ^†^
*p*-values for trend were calculated using regression analysis by inserting an interaction term for use of dietary supplements and serum retinol and α-tocopherol in the analysis. ^‡^ Multivariate logistic regression analysis was used with adjustment for age, residence, education, smoking status, alcohol consumption, physical activity, BMI, GFR, systolic and diastolic blood pressure, and log transformed hs-CRP serum retinol and α-tocopherol levels mutually.

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
