# Peer review of "Associations of Serum Retinol and α-Tocopherol Levels with Uric Acid Concentrations: Analysis of a Population-Based, Nationally Representative Sample"

_nutrients, 2020, doi:10.3390/nu12061797_

Round 1

Reviewer 1 Report

Paper is interesting and well written, however some revision is needed:

  • In the introduction also the role of uric acid in cardiovascular diseases and events should be mentioned. I will suggest the following as a possible reference since it is very recent and summarize the work of a big database study relative to cardiovascular disease in uric acid (High Blood Press Cardiovasc Prev. 2020 Apr;27(2):121-128.).
  • Please clarify wich blood pressure is used in the model 3: SBP? DBP? or MBP? the results should be similar but it need to be clarified.
  • A difference between males and females exist, in fact males loses significant for tocopherol in medol 3 while it happens in model 4 for females. It seems to indicate that in males metabolic disarrengement associated to uric acid is more strong and influent on it's level. Please discuss in the relative section.

Reviewer 2 Report

The study by Kim Y et al et al was performed to explore the independent associations of retinol and α-tocopherol with serum uric acid levels in a Korean population. 6023 participants aged ≥19 years from the Korean National Health and Nutrition Examination Survey (KNHANES) were included. Serum retinol levels were positively associated with uric acid concentrations in a dose-dependent fashion in both sexes (P <0.001). Moreover, in the multivariable logistic model, the hyperuricemia risk increased linearly with the increase in serum retinol level, regardless of sex (P <0.001). Although the serum α-tocopherol level appeared to be significantly associated with increased uric acid levels, this association was not maintained after adjusting for serum retinol levels. Authors suggest that having serum retinol concentrations under subtoxic levels might be necessary to prevent hyperuricemia and related adverse health outcomes.

The study has been well-performed and gives clear-cut results.

Introduction could give some more information about uric acid levels concerning cardiovascular risk, metabolic syndrome, and non-dipping blood pressure

Reviewer 3 Report

In this paper the authors investigate the relationship between serum retinol and α-tocopherol levels with uric acid concentrations. This is a population-based study, and the number of enrolled subjects is the major strength. However, some issues need to be clarified:

  1. Uric acid also inhibits vitamin D production and thereby results in hyperparathyroidism, I suggest the correlation in this study should be adjusted more such as serum vitamin D and PTH levels. Only serum retinol and α-tocopherol levels could be not satisfactory.
  2. Hyperuricemia is associated with many adverse health outcomes. However, monitoring the serum uric acid level would be more practical than serum retinol level in clinical practice. The availability and price of checking serum uric acid are much better than checking serum retinol.
  3. Many drugs and different food can make the different effects on the serum and urine uric levels, how the authors make the corrections between them?

Round 2

Reviewer 3 Report

Congratulations on a well written manuscript.
This revised manuscript now is well written focus on association among serum retinol, α-tocopherol levels, and uric acid concentrations. The sample size is quite large and authors have used rigorous statistical methods. I have no questions again.